# Excitation and inhibition onto central courtship neurons biases *Drosophila* mate choice

Benjamin R Kallman[1,2], Heesoo Kim[1,2], Kristin Scott[1,2]*

[1]Department of Molecular and Cell Biology, University of California, Berkeley, Berkeley, United States; [2]Helen Wills Neuroscience Institute, University of California, Berkeley, Berkeley, United States

**Abstract** The ability to distinguish males from females is essential for productive mate selection and species propagation. Recent studies in *Drosophila* have identified different classes of contact chemosensory neurons that detect female or male pheromones and influence courtship decisions. Here, we examine central neural pathways in the male brain that process female and male pheromones using anatomical, calcium imaging, optogenetic, and behavioral studies. We find that sensory neurons that detect female pheromones, but not male pheromones, activate a novel class of neurons in the ventral nerve cord to cause activation of P1 neurons, male-specific command neurons that trigger courtship. In addition, sensory neurons that detect male pheromones, as well as those that detect female pheromones, activate central mAL neurons to inhibit P1. These studies demonstrate that the balance of excitatory and inhibitory drives onto central courtship-promoting neurons controls mating decisions.

*For correspondence: kscott@berkeley.edu

**Competing interests:** The authors declare that no competing interests exist.

## Introduction

Across the animal kingdom, the ability to distinguish males from females is critical to select among potential mates. The specificity of mating decisions is exemplified by the *Drosophila* courtship ritual, in which males follow, sing to, and copulate with females but not males. Although much progress has been made in identifying the circuits that underlie mating decisions in the male fly brain, the sensory neurons that detect sex-specific cues and the pathways that they activate to generate sex-specific behaviors are incompletely understood.

A major advance in elucidating the neural circuits that govern male mating decisions has come from the discovery that a male-specific splice form of the Fruitless trancriptional regulator (FruM) is expressed in peripheral and central neurons that drive courtship behavior (*Manoli et al., 2005*; *Stockinger et al., 2005*), arguing that FruM marks neural circuits for courtship. Studies of the function of FruM-positive neurons has led to the identification of olfactory and gustatory neurons that detect pheromones, as well as central neurons that drive behavioral subprograms of courtship (*Datta et al., 2008*; *Ha and Smith, 2006*; *Kurtovic et al., 2007*; *Lu et al., 2012*; *Ruta et al., 2010*; *Thistle et al., 2012*; *Toda et al., 2012*; *von Philipsborn et al., 2011*).

One set of neurons that has emerged as a central driver of male mating behavior is the group of P1 (a.k.a. pMPe, pMP4 or pC1) neurons in the protocerebrum (*Cachero et al., 2010*; *Kimura et al., 2008*; *Lee et al., 2002*; *Yu et al., 2010*). Inducible activation of these neurons leads to sustained male courtship behaviors (*Inagaki et al., 2014*; *Kohatsu et al., 2011*; *Pan et al., 2012*; *von Philipsborn et al., 2011*). Moreover, these neurons are activated by female pheromones and this activation is partially inhibited in the presence of the male inhibitory pheromone, cis vaccenyl actate (cVA) (*Kohatsu et al., 2011*). These data show that P1 neurons receive sensory cues signaling

**eLife digest** Courtship displays are seen throughout the animal kingdom. For example, male birds-of-paradise are perhaps best known for the elaborate dances they use to attract a mate. Male fruit flies, belonging to the species *Drosophila melanogaster,* also perform courtship toward female flies. However, male flies do not court other males. Previous studies have shown that sex-specific chemical signals (or pheromones) are important cues that males use to direct courtship towards females. Researchers have previously identified two sets of sensory neurons that detect pheromones: one set detects female pheromones and promotes courtship, while the other detects male pheromones and inhibits courtship. However it was unclear how these sensory neurons controlled courtship behavior.

Now, Kallman et al. have studied the circuits of neurons in the fruit fly that promote or inhibit courtship when a fly detects a pheromone. The experiments identified several pathways of neurons in the brain of male *Drosophila* that respond to female and male pheromones. These pathways send signals that either excite or inhibit a central target, called P1 neurons. Female pheromones activated a pathway that activates the P1 neurons, whereas male pheromones activate another pathway that inhibits the P1 neurons. Kallman et al. suggest that the balance of these excitatory and inhibitory signals controls a fly's decision to court.

Following on from this work one of the next challenges will be to identify the neural circuits that act downstream of the P1 neurons to control courtship. Future studies could also explore how P1 neurons integrate signals from different senses.

females or males and drive courtship decisions, suggesting that they may be command neurons for courtship behaviors.

The sensory pathways that converge onto P1 neurons are poorly defined. Diverse sensory stimuli contribute to courtship decisions, including visual, auditory, and chemosensory cues. Important sensory cues detected primarily by contact chemosensory neurons are sex-specific cuticular hydrocarbons that act as pheromones. Multiple gustatory receptors and neurons have been implicated in pheromone detection (*Bray and Amrein, 2003*; *Koh et al., 2014*; *Miyamoto and Amrein, 2008*; *Moon et al., 2009*; *Watanabe et al., 2011*). We and others recently showed that leg chemosensory neurons expressing the PPK23 pickpocket ion channel detect pheromones (*Lu et al., 2012*; *Thistle et al., 2012*; *Toda et al., 2012*). PPK23 is expressed in sensory neurons of many leg chemosensory bristles, with generally two PPK23 cells per bristle. One cell responds selectively to male pheromones (M cells) and the other cell to female pheromones (F cells) (*Pikielny, 2012*; *Thistle et al., 2012*). In contrast, the PPK25 channel is expressed in one of the two PPK23-positive cells per bristle, and PPK25 is required for cellular and behavioral responses to female pheromones, arguing that it selectively labels F cells (*Starostina et al., 2012*; *Vijayan et al., 2014*). Unlike other classes of gustatory neurons implicated in pheromone detection, PPK23 cells are Fruitless-positive (*Lu et al., 2012*; *Thistle et al., 2012*; *Toda et al., 2012*). This suggested that it may be possible to trace pheromonal pathways from PPK23 cells in the periphery to the central nervous system by using FruM neurons as a guide.

Here, we examine sensory pathways in the male brain, from pheromone-sensing cells on the legs to the ventral nerve cord to the protocerebrum, in order to elucidate the neural circuits that allow the male fly to distinguish between appropriate and inappropriate mates. These studies define sensory pathways that act as excitatory and inhibitory drives onto P1, providing insight into the functional connectivity of the courtship circuit.

## Results

### Genetic access to sensory neurons that detect female or male pheromones

To examine pathways activated by female excitatory pheromones and male inhibitory pheromones, we focused on different subpopulations of PPK23 cells as specific sensory inputs. By GCaMP6s

calcium imaging (*Chen et al., 2013*) of PPK23 cells in a background in which PPK25 cells were independently labeled, we first confirmed that the PPK25-positive cells (F cells) are tuned to female pheromones and the PPK25-negative cells (M cells) to male pheromones (*Figure 1A,B*; *Table 1* contains genotypes of flies used for all experiments). In addition, we found that F cells are the only leg neurons that express the *vesicular glutamate transporter-Gal4* driver (*vGlut-Gal4*) (*Daniels et al., 2008*), suggesting that the two classes differ in their neurotransmitter profiles and providing an additional marker that selectively labels F cells (*Figure 1C,D*). F cells and M cells also differ in their axonal projection patterns: F cells terminate in the ventral nerve cord (VNC) whereas M cells also have fibers that project to the subesophageal zone (SEZ) of the central brain (*Figure 1E,G*).

To ask whether F or M cell activation is sufficient to modify courtship behavior, we used genetic strategies to express the heat-activated cation channel dTRPA1 (*Hamada et al., 2008*) selectively in F or M cells. A single male was placed in a chamber with a virgin female and number of single wing extensions by the male was monitored, as this motor subprogram occurs specifically during courtship song production. Males expressing dTRPA1 in F cells tested at 30°C (a temperature that activates dTRPA1) had a significantly higher wing extension rate than genetically identical flies tested at the non-activating temperature of 23°C or control flies at either temperature (*Figure 1F*). In contrast, males expressing dTRPA1 in M cells had a significantly lower wing extension rate at 30°C compared to genetic and temperature controls (*Figure 1H*). Thus, F and M cells comprise genetically and anatomically distinct chemosensory neuron classes that respond to female or male pheromones and promote or inhibit courtship.

## F cell stimulation activates P1 neurons

The ability to genetically access and specifically activate cells responding to female or male pheromones provided the opportunity to trace sex-selective pathways in the male brain and examine the neural substrates for courtship decisions. We genetically accessed F cells with the *ppk25-Gal4*, *vGlut-LexA* and *vGlut-QF2* drivers, all of which specifically label F cells in the legs (*Figure 1C,D*, and *Figure 1—figure supplement 1*). To selectively access M cells (PPK23-positive, PPK25-negative cells), we used a driver that labels both F and M cells (*ppk23-LexA*) while driving Gal80 with F cell drivers (*Figure 1—figure supplement 1*). The ATP-gated cation channel P2X2 (*Lima and Miesenbock, 2005*) was selectively expressed in these sensory classes and ATP was applied to the legs for robust cell-specific activation.

Male-specific, Fru-positive P1 neurons (*Cachero et al., 2010*; *Kimura et al., 2008*; *Lee et al., 2002*; *Yu et al., 2010*) are prominent courtship-promoting neurons in the protocerebrum that trigger sustained courtship behaviors upon ectopic activation (*Inagaki et al., 2014*; *Kohatsu et al., 2011*; *Pan et al., 2012*; *von Philipsborn et al., 2011*). As P1 neurons have been shown to respond to hydrocarbon extracts from female and male abdomens (*Kohatsu et al., 2011*), we tested whether F and M cells provided specific sensory inputs onto P1 neurons. P1 activity was monitored by GCaMP6s calcium imaging in live flies expressing P2X2 in both F and M cells (F+M), F cells, or M cells while ATP was applied to the legs (*Figure 2A,B*). Activation of F cells triggered calcium increases in P1, demonstrating that sensory neurons that detect female pheromones activate courtship-promoting P1 neurons. In contrast, we observed no significant calcium response in P1 neurons upon M cell activation. Activating F+M cells using the same *ppk23-LexA* driver (without expression of Gal80 in F cells) caused robust calcium responses in P1 neurons. These experiments argue that F cells, but not M cells, activate P1 neurons. This is in contrast to a previous study that observed calcium increases in P1 neurons in response to male abdomens (*Kohatsu et al., 2011*); however, male abdomens may activate other sensory neurons in addition to M cells, such as Fru-negative pheromone-sensing neurons, gustatory neurons, or mechanosensory neurons. The selective activation of single classes of sensory cells allows us to disambiguate sensory cues and determine the contribution of specific sensory inputs.

## PPN1 neurons respond to F cell stimulation and activate P1 neurons

As F cells terminating in the VNC do not directly contact P1, we screened existing Gal4 collections (*Gohl et al., 2011*; *Jenett et al., 2012*) for neurons that might contact F cell projections and project to higher brain regions. We identified a pair of neurons with dendrites in the VNC and axons in the protocerebrum, marked specifically by *R56C09-Gal4*, which we name Pheromone Projection Neuron

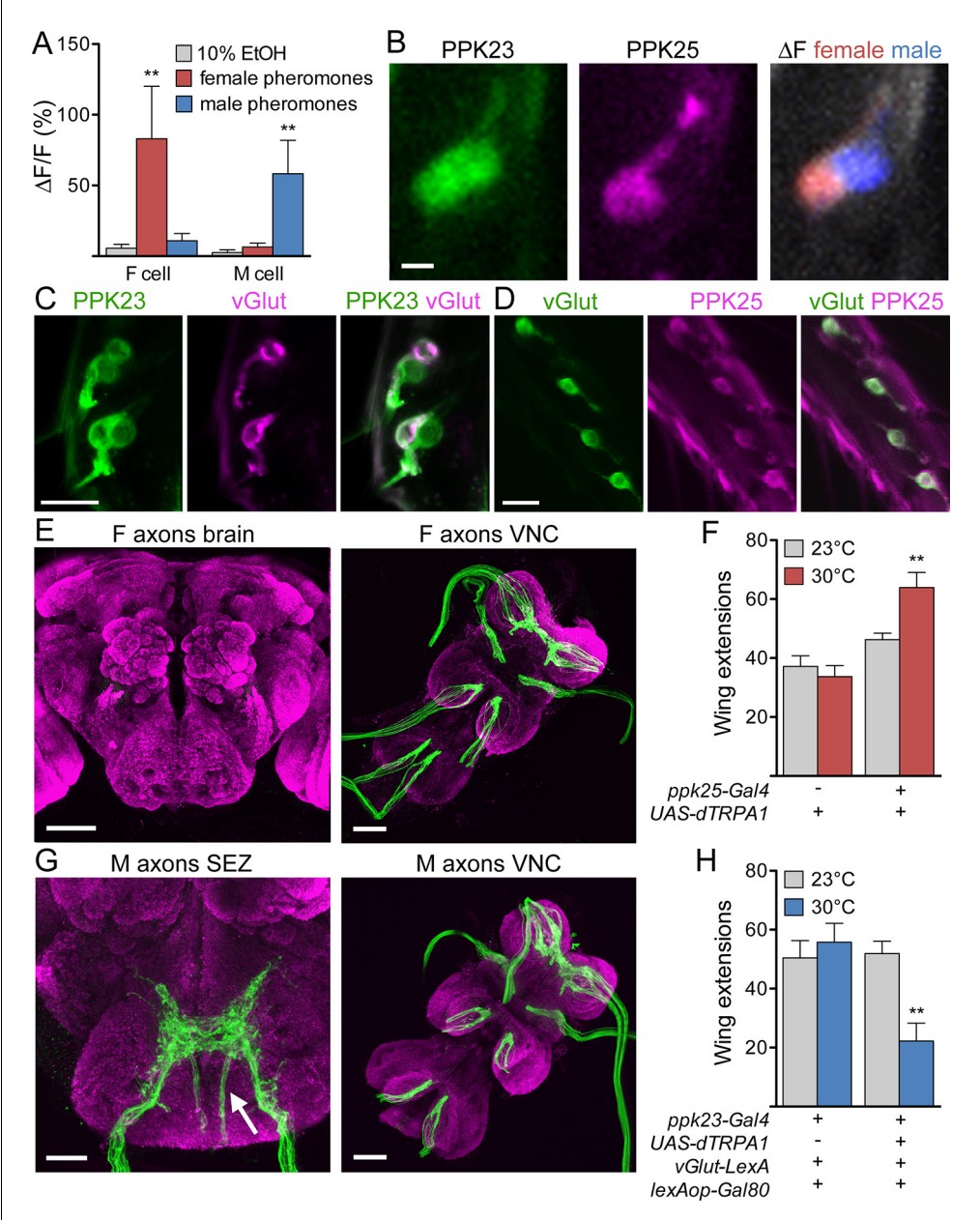

**Figure 1.** F and M cells comprise distinct chemosensory neuron classes. (**A**) F cells (PPK23+ PPK25+) respond to female pheromones whereas M cells (PPK23+ PPK25-) respond to male pheromones; n = 7 bristles. The female pheromone mix contained 7,11-heptacosadiene and 7,11-nonacosadiene. The male pheromone mix contained 7-tricosene and cis-vaccenyl acetate. (**B**) GCaMP6s marks two PPK23 cells per bristle (left). CD8::tdTomato marks the PPK25 cell (middle). Maximum $\Delta F$ of both PPK23 cells to female (red) or male pheromones (blue) (right). (**C**) vGlut (magenta) is expressed in one PPK23 cell (green) under each bristle. Transgenic flies with *ppk23-LexA, lexAop-CD2::GFP, vGlut-Gal4, UAS-CD8::tdTomato* were used for cell labeling. (**D**) vGlut (green) and PPK25 (magenta) are expressed in the same cell under each bristle. Flies contained *vGlut-LexA::VP16, lexAop-CD2::GFP, ppk25-Gal4, UAS-CD8::tdTomato*. (**E**) Axons from F cells in the legs (green) do not project to the central brain (left) but instead terminate in the six leg neuromeres of the ventral nerve cord (VNC, right). Brains are counterstained with nc82 (magenta) to show neuropil. (**F**) Expression of dTRPA1 in F cells promoted male-female courtship upon heat-evoked neural activation; n = 11–21/condition. Number of unilateral wing extensions per 10-minute trial was recorded. (**G**) M cells from the legs project to the SEZ in the brain (arrow shows fibers entering from the cervical connective) and VNC. Other SEZ axons come from the proboscis, with fibers entering from the labellar nerve. (**H**) dTRPA1-mediated activation of M cells suppressed male-female courtship. n = 10–12/condition. Number of unilateral wing extensions per 10-minute trial was recorded. Scale bars, 5 (B), 10, (C, D) 25 (G, SEZ) or 50 μm (E, G, VNC). Data are Mean ± SEM. Kruskal-Wallis test, Dunn's post-hoc (**A**) or 2-way ANOVA, Bonferroni post-hoc (**F, H**). **p< 0.01. See also *Figure 1—figure supplement 1*, on selectively targeting F or M cells.

*Figure 1. continued on next page*

*Figure 1. Continued*

The following figure supplements are available for Figure 1:

**Figure supplement 1.** Approach to selectively target F or M cells.

class 1 (PPN1) (*Figure 3A–C*). PPN1 neurons have cell bodies on the dorsal surface of the third leg neuromere, send projections to all leg and wing neuromeres in the VNC, and terminate in the ventrolateral protocerebrum. PPN1 dendrites are in close proximity to PPK25 axons and PPN1 axons overlap with P1 fibers, as shown by double labeling experiments (*Figure 3E,F*). To test whether PPN1 is involved in courtship behavior, we expressed dTRPA1 in PPN1 neurons and monitored male courtship towards females upon heat-induced neural activation. Consistent with a role in promoting courtship, activating PPN1 with dTRPA1 increased male courtship behavior toward females, as measured by unilateral wing extension rate (*Figure 3D*). Unlike other neurons of the courtship circuit, PPN1 is not Fru-positive based on intersectional approaches with *Fru-Flp* and *Fru-LexA* and PPN1 projections are not sexually dimorphic (data not shown). Nonetheless, the anatomical and behavioral studies suggest that PPN1 might transmit F cell activation to P1 to promote courtship.

To test whether PPN1 receives pheromonal signals, we stimulated F cells or M cells by ATP-mediated activation of P2X2 while monitoring calcium changes in PPN1 axons in the higher brain. These studies revealed calcium increases in PPN1 axons upon F cell stimulation but not M cell stimulation (*Figure 4A*). As with P1, activating F+M cells using the same *ppk23-LexA* driver (without expression of Gal80 in F cells) triggered robust calcium responses in PPN1, further arguing that PPN1 neurons are downstream of F cells and not M cells. To more directly test whether F cell activity is transmitted to P1 by PPN1, we expressed the red-shifted opsin Chrimson (*Klapoetke et al., 2014*) in PPN1 and monitored calcium changes in P1. Activation of PPN1 with red light generated calcium responses in P1 (*Figure 4B*). Red light had no effect on P1 activity in control animals not fed the essential cofactor retinal. Together, these experiments argue that female pheromones activate a neural pathway from F cells to PPN1 to P1 to drive courtship behavior.

## M cells and F cells activate mAL neurons

As M cell stimulation did not activate P1, we searched for other targets of F and M cells by monitoring calcium responses in all Fru neurons upon F+M cell stimulation, observing activity throughout the brain using spinning disk confocal microscopy. One set of neurons was prominently activated by F+M stimulation (*Figure 5A*). These were anatomically identifiable as mAL neurons (a.k.a. aDT2, aDTb) by their distinct arborization patterns (*Cachero et al., 2010*; *Ito et al., 2012*; *Kimura et al., 2005*; *Yu et al., 2010*). Based on their anatomy and neurotransmitter profile, mAL neurons have been proposed to be sexually dimorphic GABAergic interneurons that convey inhibitory courtship signals to the higher brain in males (*Koganezawa et al., 2010*). However, the behavioral role of mAL neurons in courtship, the sensory stimuli that activate mAL, and the relationship between mAL and other components of the courtship circuit have not been determined. To examine these questions, we visually screened Gal4 lines (*Jenett et al., 2012*) and identified *R43D01-Gal4*, which includes Fru-positive, GABAergic mAL neurons (*Figure 5B* and *Figure 5—figure supplement 1*). Consistent with the notion that mAL might be downstream of pheromone-sensing sensory neurons, we found that M cell axons and mAL dendrites overlap in the SEZ (*Figure 5—figure supplement 1*).

To test whether mAL neurons participate in courtship behavior, we conditionally activated or inactivated them using the genetic intersection of *R43D01-Gal4* and *Fru-LexA* and monitored courtship behavior. Activation of mAL neurons with dTRPA1 greatly suppressed courtship toward females compared to controls (*Figure 5C*), as measured by unilateral wing extension rate. Inactivation of mAL neurons with tetanus toxin caused robust male-male chaining, a behavior in which three or more males serially court each other (*Figure 5D*), and which was almost never observed in control animals. Furthermore, expression of RNAi against the vesicular GABA transporter (vGAT) using *R43D01-Gal4* also caused male-male chaining (*Figure 5D*), arguing that GABA release from mAL inhibits courtship. Thus, activation of mAL inhibits courtship, whereas inactivation enhances courtship, demonstrating an inhibitory role for mAL in courtship decisions.

Finally, we tested the specificity of the response of mAL neurons to pheromone sensory cell stimulation, using the *R43D01-Gal4* line to express GCaMP6s and the same drivers used for the PPN1

**Table 1.** Genotypes of flies used for experiments in this study.

| Figure panel | Genotype |
| --- | --- |
| *Figure 1A* | *w-/y; ppk23-LexA/ppk25-Gal4; lexAop-GCaMP6S/UAS-CD8::tdTomato* |
| *Figure 1B* | same as *Figure 1A* |
| *Figure 1C* | *w-/y; ppk23-LexA/vGlut-Gal4; lexAop-CD2::GFP/UAS-CD8::tdTomato* |
| *Figure 1D* | *w-/y; vGlut^{MI04979}-LexA::QFAD/ppk25-Gal4; lexAop-CD2::GFP/UAS-CD8::tdTomato* |
| *Figure 1E* | *w-/y; UAS-CD8::GFP/+; ppk25-Gal4/+* |
| *Figure 1F* | see Figure |
| *Figure 1G* | *w-/y; ppk23-LexA/ppk25-Gal4; lexAop-CD2::GFP/UAS-DTI* |
| *Figure 1H* | see Figure |
| *Figure 1—figure supplement 1A* | *w-/y; vGlut^{MI04979}-QF2/ppk25-Gal4; QUAS-mCD8::GFP/UAS-CD8::tdTomato* |
| *Figure 1—figure supplement 1B* | (left) *w-/y; ppk23-LexA/ppk25-Gal4; lexAop-CD2::GFP/UAS-DTI*<br>(middle) *w-/y; vGlut^{MI04979}-LexA::QFAD/ppk23-Gal4; QUAS-Gal80/UAS-CD8::tdTomato*<br>(right) *w-/y; vGlut^{MI04979}-QF2/ppk23-LexA; QUAS-Gal80/lexAop-CD2::GFP* |
| *Figure 2A* | *w-/y; P1-Gal4-AD/UAS-CD8::GFP;P1-Gal4-DBD/+* |
| *Figure 2B* | F cell stim: *w-/y; vGlut^{MI04979}-LexA::QFAD/lexAop-P2X2; R71G01-Gal4/UAS-GCaMP6S*<br>M cell stim: *w-/y; ppk23-LexA, lexAop-P2X2/ vGlut^{MI04979}-QF2; R71G01-Gal4, UAS-GCaMP6S/QUAS-Gal80* |
| *Figure 3A* | *w-/y; UAS-CD8::GFP/+; R56C09-Gal4/+* |
| *Figure 3B–C* | *w-/y; +/+; R56C09-Gal4/UAS-DenMark, UAS-synaptotagmin-GFP* |
| *Figure 3D* | see Figure |
| *Figure 3E* | *w-/y; R56C09-LexA/ppk25-Gal4; lexAop-CD2::GFP/UAS-CD8::tdTomato* |
| *Figure 3F* | *w-, UAS-CD8::tdTomato/y; P1-Gal4-AD/R56C09-LexA; P1-Gal4-DBD/lexAop-CD2::GFP* |
| *Figure 4A* | *w-/y; UAS-CD8::GFP/+; R56C09-Gal4/+* |
| *Figure 4B* | F cell stim: *UAS-CD8::tdTomato/y; vGlut^{MI04979}-LexA::QFAD/lexAop-P2X2; R56C09-Gal4/UAS-GCaMP6S*<br>M cell stim: *UAS-CD8::tdTomato/y; ppk23-LexA, lexAop-P2X2/ vGlut^{MI04979}-QF2; R56C09-Gal4, UAS-GCaMP6S/QUAS-Gal80*<br>F+M cell stim: *UAS-CD8::tdTomato/y; ppk23-LexA/lexAop-P2X2; R56C09-Gal4/UAS-GCaMP6S* |
| *Figure 4C* | *w-, UAS-CD8::tdTomato/y; P1-Gal4-AD/R56C09-LexA; P1-Gal4-DBD/lexAop-CD2::GFP* |
| *Figure 4D* | *w-/y; UAS-GCaMP6S/R56C09-LexA; R71G01-Gal4/lexAop-Chrimson* |
| *Figure 5A* | *w-/y; ppk23-LexA/lexAop-P2X2; fru-Gal4/UAS-GCaMP6S* |
| *Figure 5B* | *w-/y; UAS>stop>CD8::GFP/lexAop-FLPL; R43D01-Gal4/fru-LexA* |
| *Figure 5C* | see Figure |
| *Figure 5D* | see Figure |
| *Figure 5E* | F cell stim: *w-/y; vGlut^{MI04979}-LexA::QFAD/lexAop-P2X2; R43D01-Gal4/UAS-GCaMP6S*<br>M cell stim: *w-/y; ppk23-LexA, lexAop-P2X2/ vGlut^{MI04979}-QF2; R43D01-Gal4, UAS-GCaMP6S/QUAS-Gal80* |
| *Figure 5F* | same as *Figure 5e* and F+M cell stim: *ppk23-LexA/lexAop-P2X2; R43D01-Gal4/UAS-GCaMP6S* |
| *Figure 5—figure supplement 1A* | *w-/y; UAS-CD8::GFP/+; R43D01-Gal4/+* |
| *Figure 5—figure supplement 1B* | *w-/y; UAS-CD8::GFP/+; R43D01-Gal4/+* |
| *Figure 5—figure supplement 1C* | *w-/y; UAS>stop>nsyb-GFP 19a/lexAop-FLPL; R43D01-Gal4/fru-LexA* |
| *Figure 5—figure supplement 1D* | *w-/y; UAS>stop>Dscam17.1-GFP 19a/lexAop-FLPL; R43D01-Gal4/fru-LexA* |

*Table 1. continued on next page*

*Table 1.* continued

| Figure panel | Genotype |
|---|---|
| *Figure 5—figure supplement 1E* | *w-/y; UAS-CD8::GFP/ppk23-LexA; R43D01-Gal4/lexAop-myr::mCherry* |
| *Figure 5—figure supplement 1F* | *w-/y; ppk23-LexA/lexAop-P2X2; R43D01-Gal4/R71G01-Gal4, UAS-GCaMP6S* |
| *Figure 5—figure supplement 1G* | *w-/y; R43D01-LexA/+; lexAop-CD2::GFP/+* |
| *Figure 6A* | *w-, UAS-CD8::tdTomato/y; P1-Gal4DBD/R43D01-LexA; P1-Gal4AD/lexAop-CD2::GFP* |
| *Figure 6B* | see figure, genotype includes *UAS-Dicer* (X) |
| *Figure 6C* | see Figure |
| *Figure 6D* | *w-/y; ppk23-LexA/lexAop-P2X2; R71G01-Gal4, UAS-GCaMP6S/R43D01-Gal4* |
| *Figure 6E* | *w-/y; R43D01-LexA/UASArcLight; R71G01-Gal4/lexAop-Chrimson* |
| *Figure 6F* | same as *Figure 6E* |

and P1 experiments to express P2X2 in M and F cells (*Figure 5E,F*). Surprisingly, both M cells and F cells activated mAL, arguing that detection of female as well as male pheromones provides an inhibitory courtship drive. As F cells additionally activated PPN1 and P1 courtship-promoting neurons, whereas M cells did not, this suggests that the balance between excitation and inhibition underlies the decision to court.

## mAL neurons inhibit P1 courtship command neurons

How do mAL neurons inhibit courtship? P1 neurons are in close proximity to mAL termini (*Figure 6A*), suggesting that they may be candidate targets of GABAergic mAL neurons. To test whether P1 neurons receive inhibitory signals, we generated flies containing an RNAi against the GABA-A receptor subunit Resistant to dieldrin (Rdl) (*Ffrench-Constant et al., 1991*) in P1 neurons and examined the behavioral consequence. Males expressing Rdl RNAi in P1 neurons displayed increased courtship toward other males, arguing that P1 neurons receive GABAergic inhibition (*Figure 6B*). In addition, whereas Chrimson-mediated activation of P1 neurons induced courtship

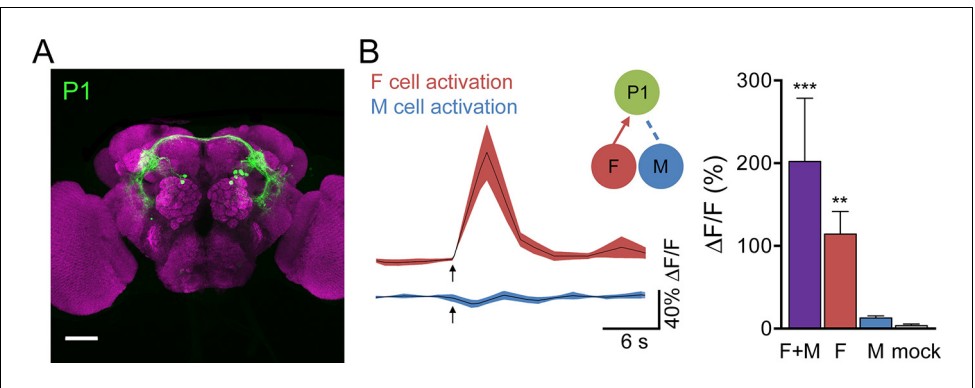

**Figure 2.** F cells activate courtship-promoting P1 neurons. (**A**) Male-specific P1 neurons (green) are located in the protocerebrum. Scale bar, 50 μm. Flies contained *P1-Gal4DBD, P1-Gal4AD, UAS-CD8::GFP*. (**B**) ATP-mediated stimulation of F+M cells (*ppk23-LexA, lexAop-P2X2*), F cells (*vGlut-LexA*) but not M cells (*ppk23-LexA, lexAop-P2X2, vGlut-QF2, QUAS-Gal80*) triggered calcium increases in P1 neurons; n = 5–8/condition. Mock is no P2X2. Traces on the left show averaged △F/F with mean in black and SEM shaded. Arrows indicate stimulus. Schematics show cells monitored with GCaMP6s (green) and connections tested. Data are also displayed as bar graph (right). Differences in expression levels of *ppk23-LexA* and *vGlut-LexA* may contribute to different response magnitudes of F+M versus F cell stimulation. Mean ± SEM of maximum △F/F. Kruskal-Wallis test, Dunn's post-hoc to mock, *p<0.05, **p<0.01, ***p<0.001.

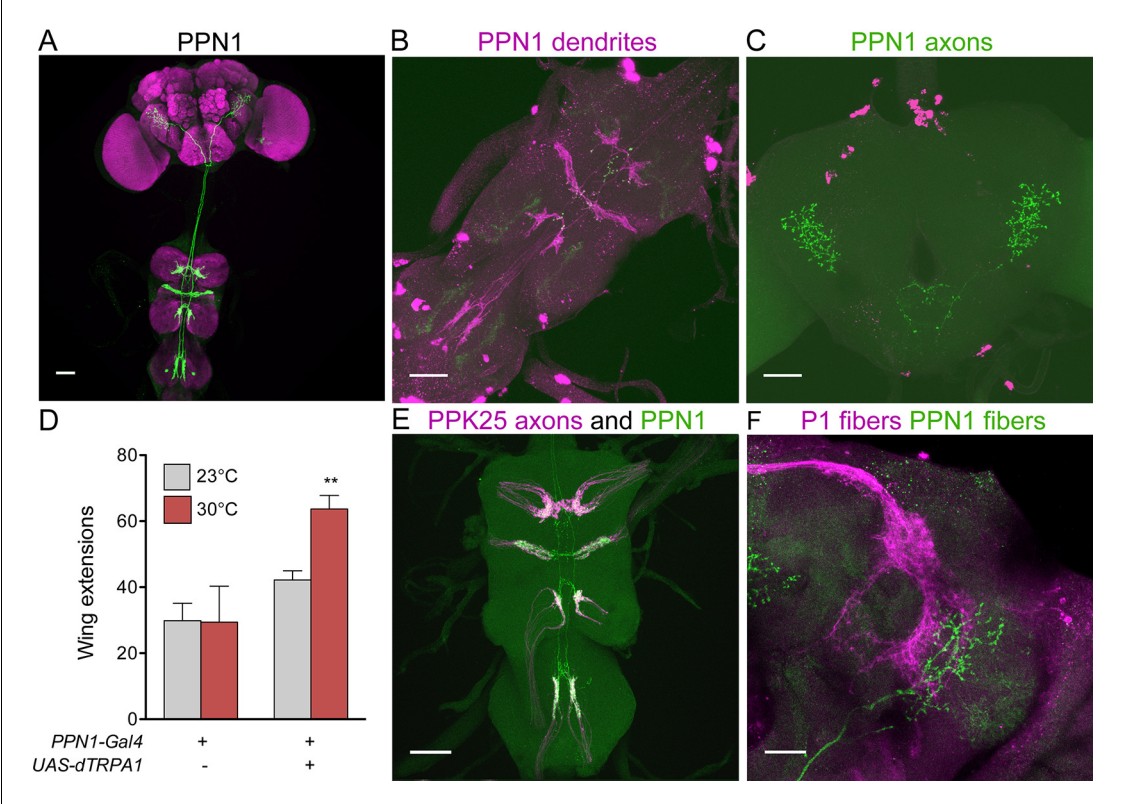

**Figure 3.** PPN1 neurons are courtship-promoting neurons in proximity to PP23 axons and P1 fibers. (A) PPN1 neurons have cell bodies in the third leg neuromere of the VNC and send projections to the six leg neuromeres and wing neuromere of the VNC and to the ventrolateral protocerebrum of the brain. *R56C09-Gal4* drives expression of *UAS-CD8::GFP* exclusively in the pair of PPN1 neurons. (B-C) PPN1 has dendrites in the VNC (B, DenMark, magenta) and axons in the ventrolateral protocerebrum (C, syt-GFP, green). B and C are from the same animal containing *R56C09-Gal4, UAS-DenMark, UAS-synaptotagmin-GFP*. (D) Activation of PPN1 with dTRPA1 causes increased male-female courtship at 30˚C; mean ± SEM, n = 16–30/condition, **p<0.01 (2-way ANOVA, Bonferroni post-hoc). (E) Overlap is observed in the VNC between PPN1 dendrites (green) and incoming PPK25 axons (magenta). *R56C09-LexA, lexAop-CD2::GFP, ppk25-Gal4, UAS-CD8::tdTomato* flies were used. F. Overlap between PPN1 (green) and P1 (magenta) in the anterior ventrolateral protocerebrum (50 µm collapsed Z-stack). Scale bars, 25 µm (F) 50 µm (A-E).

behavior in solitary males, co-activation of P1 and *R43D01-Gal4* neurons suppressed this behavior (*Figure 6C*), arguing that mAL suppresses P1-mediated courtship. To more directly test whether inhibitory signals from mAL impinge on P1, we simultaneously stimulated F and M cells and monitored activity in P1 before and after 2-photon guided lesioning of mAL axons (*Figure 6D* and *Figure 5—figure supplement 1*). Indeed, simultaneous stimulation of F and M cells caused transient activation of P1 and lesioning mAL axons significantly increased P1 activation.

These data suggest that mAL neurons inhibit P1 neurons via GABA-A receptors. To test directly whether mAL input onto P1 causes hyperpolarization, we expressed ArcLight (*Cao et al., 2013*), a fluorescent voltage sensor, in P1 neurons and monitored its fluorescence using 2-photon imaging (*Figure 6E*). Chrimson-mediated activation of mAL neurons, using *R43D01-LexA* (*Figure 5—figure supplement 1*), evoked a rapid increase in ArcLight fluorescence in P1 neurons, indicating hyperpolarization. No fluorescence change was observed in control flies lacking the *R43D01-LexA* transgene. These experiments demonstrate that mAL neurons provide an inhibitory drive onto P1 neurons.

## Discussion

This work identifies pheromone-responsive neural circuits underlying mating decisions (*Figure 7*). We used cell-specific activation to determine sensory pathways that impinge on courtship-promoting P1 neurons, providing insight into the functional connectivity of the courtship circuit. Female pheromones trigger neural pathways that excite and inhibit P1, with PPN1 providing an excitatory drive

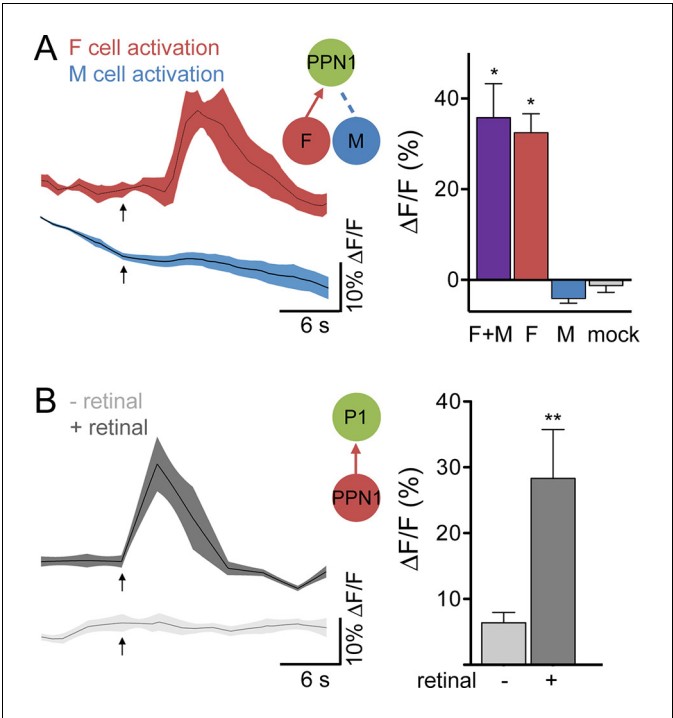

**Figure 4.** F cells activate courtship-promoting PPN1 neurons which activate P1. (**A**) Calcium imaging of PPN1 axons while activating different sensory classes revealed that PPN1 is activated by F+M and F cell stimulation but not M cell stimulation; n = 5–7/condition. Mock is no P2X2. Arrows indicate stimulus. (**B**) Chrimson-mediated activation of PPN1 triggers calcium increases in P1 by GCaMP6s calcium imaging; n = 5–6/condition. Arrows indicate stimulus. Schematics show cells monitored with GCaMP6s (green) and connections tested. Data are Mean ± SEM. Kruskal-Wallis test, Dunn's post-hoc to mock (A) or Mann-Whitney test (B). *p<0.05, **p<0.01.

onto P1 and mAL providing an inhibitory drive. F cell activation leads to P1 activation and increased courtship behavior, arguing that the sum of inputs onto P1 produces excitation or that the sequence of inputs, i.e., fast excitation followed by inhibition, provides a temporal window for excitation. In contrast, M cells activate mAL neurons to inhibit P1 and inhibit courtship behavior. These studies argue that the balance of excitation and inhibition onto P1 neurons is different following F cell or M cell activation: F cell activation leads to overall P1 activation whereas M cell activation leads to overall P1 inhibition.

Our calcium imaging studies provide strong support that the pathways identified in this study are activated by F and M cell stimulation, and our behavioral experiments demonstrate that activation of these neurons directly contributes to courtship decisions. However, we do not exclude the possibility that additional intermediary neurons may also be activated by F or M cells. Indeed, a recent study identified vAB3 neurons, with projections in the first leg neuromere in the VNC, the SEZ and protocerebrum, as activated by female but not male pheromones (*Clowney et al., 2015*). This study also showed that mAL neurons are activated by male and female abdomens and provided evidence that mAL inhibits P1. Our independent observations are consistent with this work and extend the findings by providing behavioral evidence that each identified component of the circuit promotes or inhibits courtship as predicted by its response properties, by using cell-type specific activation strategies coupled with cell-type specific imaging studies to directly test potential connections, and by identifying PPN1 as a novel neural component of the courtship circuit. Together, the studies argue that pheromones activate multiple excitatory and inhibitory interneurons that impinge on P1.

A caveat of our studies is that they rely on ectopic expression of reporters, and expression levels of P2X2 and GCaMP6s may influence the ability to detect responding cells. Nevertheless, the same driver was used to express P2X2 in M cells in all calcium-imaging experiments, but application of ATP only elicited calcium responses in mAL neurons. Furthermore, the *ppk23-LexA* driver (in F+M

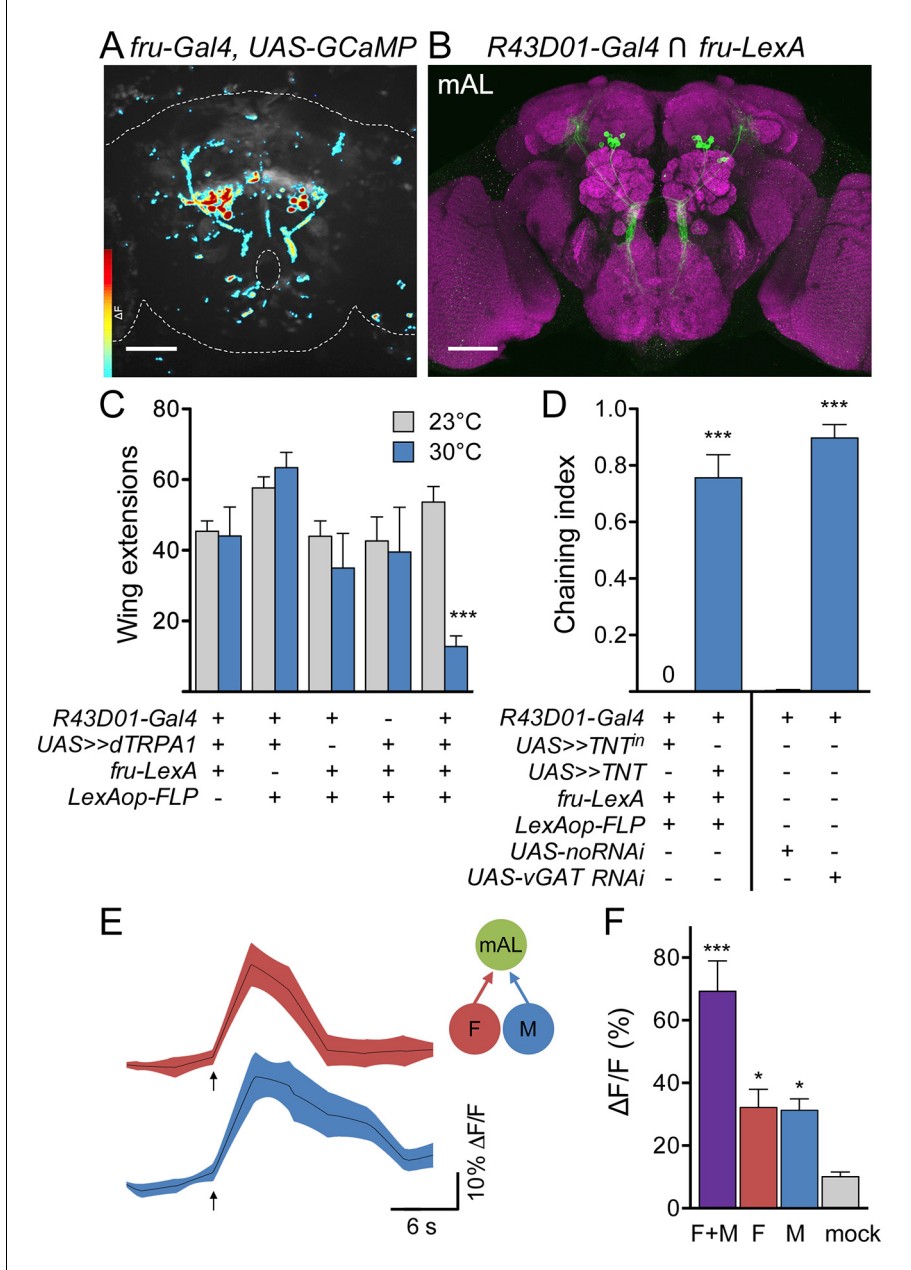

**Figure 5.** M cells and F cells activate courtship-suppressing mAL neurons. (**A**) Example image of a GCaMP6s ΔF heat map in *fru-LexA* neurons upon P2X2-mediated activation of F+M cells. (**B**) mAL neurons labeled by the intersection of *R43D01-Gal4* and *fru-LexA* connect the SEZ and protocerebrum. (**C**) Activating mAL neurons with dTRPA1 suppresses courtship toward females; n = 10/condition. (**D**) Silencing mAL neurons with tetanus toxin or knocking down vGAT with *R43D01-Gal4* induces male-male chaining; n = 8–10 groups/condition, 6–9 males per group. For C and D, >> means >stop>. Chaining index represents the fraction of time 3 or more males were courting over the 10-min trial. (**E**) P2X2-mediated stimulation of either F or M cells activates mAL neurons by GCaMP calcium imaging. Arrows indicate stimulus. (**F**) Maximum ΔF/F in mAL cell bodies; n = 5–9/condition. Mock is no P2X2. Scale bars, 50 μm (A, B). Data are Mean ± SEM, 2-way ANOVA, Bonferroni post-hoc (C), Mann-Whitney test (D), or Kruskal-Wallis test, Dunn's post-hoc to mock (F). *p<0.05, ***p<0.001. See also *Figure 5—figure supplement 1*, for anatomical characterization of mAL neurons.

The following figure supplements are available for Figure 5:

**Figure supplement 1.** Characterization of mAL neurons.

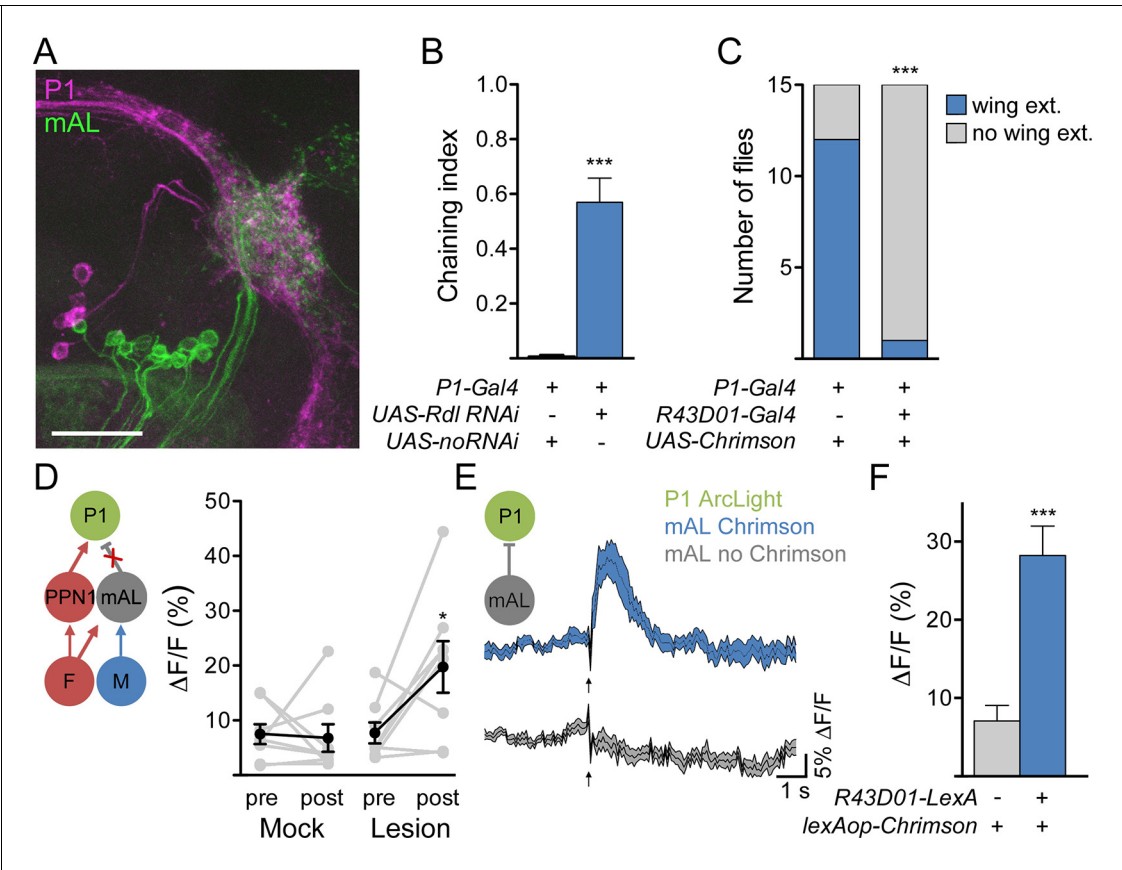

**Figure 6.** mAL neurons functionally and behaviorally inhibit P1 neurons. (**A**) Overlap between mAL (green) and P1 (magenta) in the superior lateral protocerebrum (collapsed 97-μm stack). Scale bar is 25 μm. (**B**) Knockdown of GABA_A receptor Rdl in P1 neurons induces male-male chaining; n = 10/condition. (**C**) Chrimson-mediated activation of P1 neurons causes wing extension (wing ext.) in solitary males, which is suppressed by co-activation of neurons expressing *R43D01-Gal4*. (**D**). Lesioning mAL axons increases GCaMP response in P1 upon P2X2-mediated stimulation of F+M cells. (**E**) Chrimson-mediated activation of mAL neurons causes P1 hyperpolarization, as detected by increased ArcLight fluorescence. Arrow indicates laser. F. Maximum $\Delta F/F$ in P1; n = 10–11/condition. Data are Mean ± SEM. Mann-Whitney test (B, F), Fisher's exact test (C), or paired t-test (D). *p<0.05, **p<0.01, ***p<0.001.

cells) that we used for M cell activation was sufficient to produce ATP-mediated activation of P1 and PPN1, and these responses were abolished in the presence of Gal80 in F cells. Thus, the observation that M cells activate mAL but not P1 or PPN1 is unlikely due to technical limitations.

This study demonstrates a specific computational logic used by the nervous system to integrate different sensory inputs. Pheromones provide excitatory or inhibitory drives onto P1, such that P1 activity reflects the integration of positive and negative sensory inputs, with female pheromones causing net excitation and male pheromones causing net inhibition. P1 also integrates inputs from other sensory systems, as P1 neurons respond to visual stimuli (*Kohatsu and Yamamoto, 2015*) and olfactory pheromones (*Clowney et al., 2015*; *Kohatsu et al., 2011*). Thus, diverse sensory stimuli may alter the weight of excitation versus inhibition onto P1 and bias the decision to court. Altering the weights of excitation or inhibition through experience or evolution is an appealing strategy to dynamically modulate the response to potential mates or to tune attraction to conspecifics. Taken together, these studies reveal an elegant strategy used by the nervous system in which excitatory and inhibitory inputs directly converge onto a common output to control a significant behavioral decision.

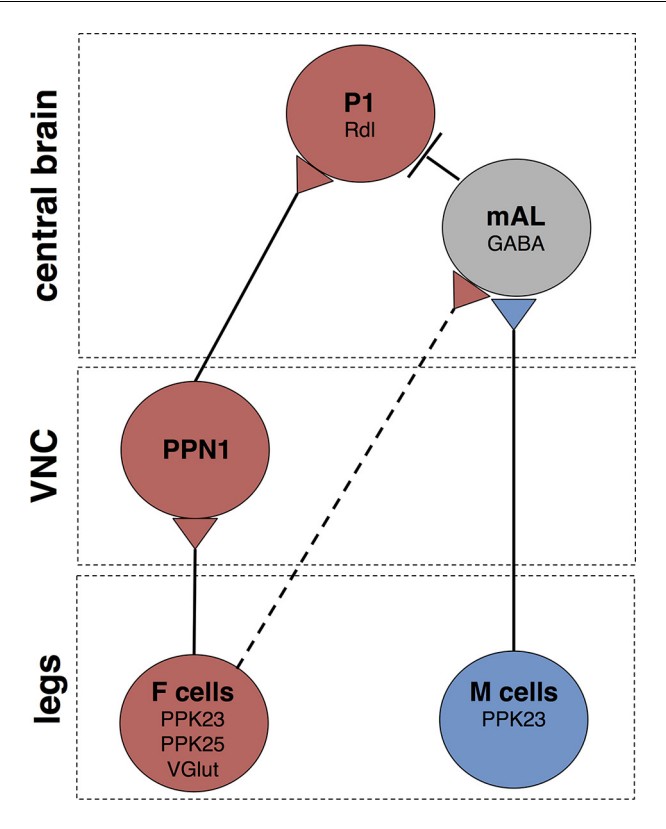

**Figure 7.** Schematic of courtship-promoting and courtship-inhibiting circuits activated by F and M cells. F cells on the leg express PPK23, PPK25, and VGlut, and respond to female pheromones. M cells on the leg express PPK23 and respond to male pheromones. The M cell neurotransmitter is unknown. F cells activate PPN1, a class of projection neuron with cell bodies and dendrites in the VNC and long-range axonal projections to the ventrolateral protocerebrum. PPN1 axons are in close proximity to P1 fibers, and PPN1 activation causes activation of P1. M cells on the leg activate GABAergic mAL neurons, which connect the SEZ and superior lateral protocerebrum. mAL axons interdigiate with P1 fibers, and mAL acitvation causes hyperpolarization of P1, likely through the GABA-A receptors containing the Rdl subunit. F cells also provide an inhibitory drive onto P1 via mAL. The contact between F cells and mAL is not direct (dotted line). Other connections may not be monosynaptic.

## Materials and methods

### Fly strains

The following fly lines were used

*ppk23-LexA* (a gift from Dr. Barry Dickson); *lexAop2-IVS-GCaMP6S-SV40 (BDSC #44273); ppk25-Gal4 (Starostina et al., 2012); ppk23-Gal4, lexAop-Gal80, UAS-mCD8::tdTomato (X II III) (Thistle et al., 2012); vGlut-Gal4 (OK371-Gal4) (Daniels et al., 2008); vGlut-LexA::VP16 (Baek et al., 2013); vGlut^{MI04979}-LexA::QFAD, vGlut^{MI04979}-QF2 (Diao et al., 2015); UAS-mCD8::GFP and lex-Aop-CD2::GFP (Lai and Lee, 2006); UAS-DTI (Han et al., 2000); fru-Gal4 (Stockinger et al., 2005); fru-LexA (Mellert et al., 2010); lexAop2-FLPL(attp40) (BDSC #55820); UAS-GCaMP6S (Chen et al., 2013); UAS-dTRPA1 (Hamada et al., 2008); UAS-syt-eGFP (Zhang et al., 2002); UAS-Denmark (Nicolai et al., 2010); UAS-CD4::GFP1-10 and lexAop-CD4::GFP11 (Gordon and Scott, 2009); UAS-P2X2 (Lima and Miesenbock, 2005); lexAop-P2X2 (Yao et al., 2012); UAS>stop>CD8::GFP, UAS>-stop>dTRPA1^{myc}, UAS>stop>TNTactive, UAS>stop>TNTinactive (von Philipsborn et al., 2011); UAS-Empty-RNAi (BDSC #36303); UAS-vGAT-shRNA (BDSC #41958); UAS-Dicer2 (X) (BDSC #24644); UAS-Dicer2(II) (BDSC #24650); UAS-Rdl-RNAi (BDSC #31286); QUAS-Gal80 (BDSC #51950); P1-Gal4 (Inagaki et al., 2014); R71G01-Gal4 (Pan et al., 2012); R43D01-Gal4 (BDSC #48151); PPN1-Gal4 (R56C09-Gal4, BDSC #39145); PPN1-LexA (R56C09-LexA, BDSC #53584); UAS-*

IVS-CsChrimson.mVenus(attp18) (BDSC #55134); LexAop2-IVS-CsChrimson.mVenus(attp2) (BDSC #55139); UAS-ArcLight (*Cao et al., 2013*); UAS>stop>nsyb-GFP 19a; UAS>stop>Dscam17.1-GFP 19a (*Yu et al., 2010*), QUAS-mCD8::GFP (BDSC #30003).

## Leg bristle GCaMP6s imaging

Single bristle imaging was performed as previously described (*Thistle et al., 2012*). Flies containing *ppk23-LexA, LexAop-GCaMP6S, ppk25-Gal4, UAS-CD8::tdTomato* were placed in a custom imaging chamber and their forelegs were secured with wax. Female (7,11-heptacosadiene and 7,11-nonacosadiene) or male (7-tricosene and cis-vaccenyl acetate) pheromone mixes (100 ng/µL, Cayman Chemical, Ann Arbor, MI) were applied to single bristles on the three distal leg segments for 30 s. GCaMP6s responses were captured using a 3i spinning disk confocal system equipped with a 20x air objective and 1.6x optical zoom. Responses in the two PPK23 cells under each bristle were analyzed based on their expression of CD8::tdTomato; M cells were tdTomato-negative, and F cells were tdTomato-positive. The change in fluorescence in each cell was calculated as follows: $100*((F_t-F_0)/F_0)$, where $F_0$ is the mean fluorescence intensity during the 4 s prior to stimulation.

The heat map in *Figure 1B* was created in Fiji. The average fluorescence intensity of the 10 frames preceding each stimulation (female and male mixes) was subtracted from the frame at which the fluorescence intensity of the responding cell was at a maximum. The resulting images from each stimulation were then merged into a two-channel image. The channels corresponding to the female or male pheromone stimulation were pseudocolored red and blue, respectively.

## Central brain GCaMP6s imaging and ATP stimulations

GCaMP imaging experiments were performed as previously described (*Harris et al., 2015*). Virgin males were collected at eclosion and were aged in isolation for 2 to 6 days before imaging. They were briefly anaesthetized with $CO_2$ and placed into a small slit on a custom-built plastic mount at the neck such that the head was isolated from the rest of the body. The head was then immobilized using nail polish. Two small pieces of plastic were affixed with nail polish to the underside of the plastic mount on either side of the thorax, such that the legs were forced into a forward-facing position. The proboscis was covered with wax to prevent labellar taste input. The head cuticle was dissected with fine forceps in ice-cold adult hemolymph-like solution (AHL) (*Wang et al., 2003*), and obscuring air sacs and other debris were removed. Eyes were damaged or removed to minimize visual input from the imaging laser. A coverglass was placed at a 45-degree angle to the plane of the plastic mount such that the head was isolated from the rest of the body.

GCaMP6s responses were captured using a fixed-stage 3i spinning disk confocal system equipped with a 20x water objective and 1.6x (mAL and P1) or 2.5x (PPN1) optical zoom and a 488 nm laser. During the stimulation, stacks of 15–20 Z-slices (1–1.5 µm/Z-slice, for mAL and P1) or 8–12 Z-slices (0.5–0.8 µm/Z-slice, for PPN1) were obtained with a 100-ms exposure per Z-slice, resulting in each imaging volume/timepoint being acquired every 1.7–3.9 s. For each trial, 20 imaging timepoints were acquired.

For P2X2-mediated stimulation of M and/or F cells, ~4 µL of 100 mM ATP (adjusted to pH 7) was pipetted onto a small cube of 2% agar, which was placed on the end of a glass capillary (OD 1.0 mm, ID 0.78 mm). The capillary was placed into an electrode holder that was secured to a micromanipulator. The agar cube was advanced such that it touched the flies' legs at timepoint 6 of 20. The cube was left within reach of the flies' legs for 3 timepoints before being removed.

F and M cells were exogenously activated to ensure specific stimulation of different sensory classes. Delivery of synthetic pheromones requires solubilization in ethanol or hexane, solvents that dissolve the fly's endogenous cuticular hydrocarbons, which may independently activate F or M cells. It was possible to apply pheromone solutions to the tip of a single sensory bristle without detriment (*Figure 1A*).

For PPN1 imaging, because GCaMP6s baseline fluorescence was very low, a red reporter (*UAS-CD8::tdTomato*) was included in order to visualize axonal endings. One stack of 15–20 Z-slices from the 561 nm laser line was obtained at the beginning of each imaging session. These were later used to define regions of interest for imaging analysis.

## Chrimson-mediated activation of PPN1

For PPN1 activation experiments, we fed isolated adult male flies for a minimum of 3 days on standard fly food supplemented with all-*trans* retinal (final concentration 400 μM). These flies, as well as control males fed normal food, were kept in constant darkness until the experiment. Imaging was performed similarly to above ("GCaMP6s imaging") except that a 635 nm laser (Laserglow, Canada) was directed at the thorax. The laser was turned to its highest power (~0.01 mW/mm$^2$) but was in standby mode until frame 6 of 20, at which time the key was turned to open the shutter. The laser was left on for 6 frames. During imaging, a 525/45 bandpass emission filter (Semrock, Rochester, NY) was used to prevent the 635 nm laser light from interfering with the GCaMP signal. In some cases, we found that the 488 nm imaging laser was sufficient to activate Chrimson and trigger calcium responses in downstream neurons (i.e., ppk23>Chrimson, P1>GCaMP6s). In addition, we found that Chrimson was weakly activated in the absence of retinal (see ArcLight imaging below). However, these phenomena were not observed in PPN1>Chrimson flies, likely due to weak PPN1 or Chrimson transgene expression.

## GCaMP imaging analysis

For *Figures 2, 4 and 6D*, calcium imaging data were processed in Fiji. For PPN1, a red anatomy scan (with 561nm laser) measuring CD8::tdTomato fluorescence was taken prior to GCaMP calcium imaging. A maximum intensity Z-projection for the anatomy scan and each GCaMP timepoint was used for analysis. The anatomy projection was used to draw an ROI covering the axonal region in the ventrolateral protocerebrum ("anatomical ROI"). A second ROI was drawn in a region lacking both tdTomato and G-CaMP signal ("background ROI"). Mean fluorescence levels from the background ROI was subtracted from the anatomical ROI at each GCaMP timepoint resulting in the fluorescence trace over time: $F_t$. $\triangle F/F(\%)$ was calculated as follows: $100*((F_t-F_0)/F_0)$, where $F_0$ is the mean fluorescence intensity during time points 2 to 5. For P1 fibers, $\triangle F/F(\%)$ was calculated in the same way, except that in place of an anatomical CD8::tdTomato scan, "anatomical ROIs" covering P1 commisural fibers were drawn using the maximum projection across time of the GCaMP signal. Maximum $\triangle F/F(\%)$ was calculated by subtracting the average $\triangle F/F(\%)$ of the 3 timepoints preceding the stimulation from the maximum $\triangle F/F(\%)$ of the 4 timepoints following the stimulation. Due to unavoidable differences in the background fluorescence between pre- and post-ablation imaging scans, the $\triangle F/F$ values presented in *Figure 6D* were calculated without background subtraction.

For *Figure 5*, calcium imaging data were processed in Matlab. ROIs were drawn around mAL cell bodies in single slices. $\triangle F/F(\%)$ was calculated as follows: $100*((F_t-F_0)/F_0)$, where $F_0$ is the mean fluorescence intensity during time points 2 to 5 and $F_t$ is the fluorescence at each timepoint. Maximum $\triangle F/F(\%)$ was calculated by subtracting the average $\triangle F/F(\%)$ of the 3 timepoints preceding the stimulation from the maximum $\triangle F/F(\%)$ of the 4 timepoints following the stimulation.

For the heatmap in *Figure 5A*, $\triangle F$ values were calculated for each pixel in each slice at each timepoint, generating a 4-dimensional data set. These data were collapsed spatially into a 3-dimensional data set using a maximum intensity projection in the Z dimension. The heat map represents the maximum $\triangle F$ values that occurred during stimulus (timepoints 6–9). This heatmap was overlaid on a grayscale image that is the maximum intensity projection of the average baseline fluorescence (timepoints 2–5). The color bar scale represents the minimum (blue) to maximum (dark red) $\triangle F$.

For all GCaMP data, averaging the $\triangle F/F(\%)$ traces across animals required re-sampling the individual $\triangle F/F(\%)$ traces at 10 Hz (completed with Matlab using a linear interpolation), due to the variable duration of timepoints between animals.

## Two-photon laser-mediated ablations

Ablations were performed on a Zeiss LSM 780 NLO AxioExaminer microscope. Flies expressed GCaMP6s in both mAL and P1 neurons, visualized using 488 nm light. A rectangular ROI (approximately 5 μm x 15 μm) was drawn to cover the width of the tract carrying mAL axons in a single z-section located in the middle of the tract. We then scanned the ROI 10 times (3.15 μs pixel dwell time) with intense 760 nm light (~50 mW at the front lens). Lesions were considered successful when the mAL axonal tract became discontinous. Mock ablations were performed identically except that the ROI was moved lateral to the mAL axonal tract (at the edge of the optic lobe). All ablations were performed bilaterally. Pre- and post-ablation stimulation of F and M cells were performed on a

spinning disk microscope as described above. We waited 10–20 min after the ablation to stimulate the fly with ATP ("post-ablation" condition).

## ArcLight imaging with mAL activation

Isolated adult male flies were fed standard fly food supplemented with all-*trans* retinal (final concentration 400 µM) for a minimum of 2 days and were kept in constant darkness until the experiment. Three- to 5-day-old flies were prepared as described above (GCaMP6s imaging). Imaging was performed on a Zeiss LSM 780 NLO AxioExaminer microscope. To find the region of greatest mAL-P1 overlap, Chrimson.mVenus in mAL neurons was briefly imaged with low-intensity 514 nm light, and the ROI to be scanned (approximately 30 µm x 30 µm) was drawn around the distal mAL axons where they interdigitate with P1 fibers ("signal ROI"). A second region of interest ("background ROI") was drawn in an area lacking ArcLight or Chrimson.mVenus fluorescence. ArcLight was excited with 925 nm light and scanned at approximately 15 Hz. To activate mAL neurons, flies were stimulated 5 times (~5 s/stimulation, ~30 s between stimulations) with a 635-nm laser (Laserglow, Canada, ~0.01 mW/mm$^2$). We observed weak responses in flies expressing Chrimson in mAL but not fed retinal. These responses were significantly smaller than the responses in flies fed retinal.

## ArcLight imaging analysis

To calculate $\triangle F/F$ of P1 ArcLight signal, the background ROI intensity trace was first subtracted from the signal ROI intensity trace, resulting in F. For each fly, the 5 laser stimulations were then aligned such that laser onset for each stimulation was t = 0 and the average was taken. $\triangle F/F(\%)$ for each animal was calculated as follows: $100*((F-F_0)/F_0)$, where $F_0$ is the mean fluorescence intensity over the period from 0.7 to 2.6 s preceding the stimulation. Maximum $\triangle F/F(\%)$ was calculated by subtracting the average $\triangle F/F(\%)$ of 2 s preceding the stimulation from the maximum $\triangle F/F(\%)$ of 2 s following the stimulation.

## Courtship behavior

Courtship behavior experiments were performed essentially as described (*Thistle et al., 2012*), with the following modifications: male-female assays were recorded for 10 min; assays involving *UAS-dTRPA1* were performed at room temperature (~23°C) and 30°C; for *Figure 3E*, due to the relative weakness of *UAS>stop>TRPA1$^{myc}$*, male flies were pre-incubated at 30°C for 5 min before being presented with a female; assays involving Chrimson were performed under white light (~0.05 mW/mm$^2$) with male flies fed 400 µM all-*trans* retinal for a minimum of 3 days in the dark. We found that bright white light was sufficient to activate Chrimson and cause behavioral phenotypes.

## Immunohistochemistry

Antibody staining and immunohistochemistry were performed as previously described (*Wang et al., 2004*). The following primary antibodies were used: rabbit anti-GFP (Invitrogen, Carlsbad, CA, 1:1,000), mouse anti-GFP (Invitrogen, Carlsbad, CA, 1:1,000), mouse anti-nc82 (Hybridoma Bank, Iowa City, IA, 1:500), rabbit anti-RFP (Clontech, Mountain View, CA, 1:500); rabbit anti-GABA (Sigma-Aldrich, St. Louis, MO, 1:1,000); rabbit anti-FruM (1:100). For GRASP experiments, we used a mouse monoclonal antibody that specifically recognizes reconstituted GFP (Sigma-Aldrich, St. Louis, MO, 1:200). Secondary antibodies were Alexa Fluor goat anti-mouse 488, goat anti-rabbit 488, goat anti-mouse 568, goat anti-rabbit 568 (Life Technologies, Carlsbad, CA, 1:100).

## Transgene generation

To generate *R43D01-LexA*, 1157 bp fragment from genomic DNA, including the entire R43D01 tile from the FlyLight collection (*Pfeiffer et al., 2012*), was amplified using the primers ttgagcacggatttcagcag and ggggtcctcaaatgtgtcgatttgt. This fragment was recombined into the pBPLexA::p65Uw plasmid (*Pfeiffer et al., 2010*), and inserted into the VK00018 landing site (*Venken et al., 2006*).

## Acknowledgements

Sandra Fendl and Alex Naka assisted with behavioral studies. Cosmos Wang assisted with anatomical studies. The Scott laboratory provided comments on the manuscript. This work was supported by a grant from the NIH, NIDCD to KS.

## Additional information

### Funding

| Funder | Grant reference number | Author |
| --- | --- | --- |
| National Institute on Deafness and Other Communication Disorders | R01 DC013280 | Kristin Scott |

The funders had no role in study design, data collection and interpretation, or the decision to submit the work for publication.

### Author contributions

BRK, Conception and design, Acquisition of data, Analysis and interpretation of data, Drafting or revising the article; HK, Acquisition of data, Analysis and interpretation of data, Drafting or revising the article; KS, Conception and design, Drafting or revising the article

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
