## [Decision Letter]

Thank you for submitting your work entitled "Excitation and inhibition onto central courtship neurons biases *Drosophila* mate choice" for consideration by *eLife*. Your article has been reviewed by two peer reviewers, and the evaluation has been overseen by Mani Ramaswami as Reviewing Editor and a Senior Editor.

The reviewers have discussed the reviews with one another and the Reviewing editor has drafted this decision to help you prepare a revised submission.

Summary:

Kallman et al. present a highly interesting study that elucidates the neuronal circuitry underlying the processing of female and male pheromones. This study is a tour de force, supported by beautiful data. The authors use a wide range of advanced methods including functional imaging, optogenetics, anatomical methods as well as behavioral assays to dissect in detail the neural pathways involved in courtship behavior. The authors show that F cells (PPK25-positive/*vGlut*-positive), which are tuned to female pheromones, promote courtship behavior, while their anatomical counterpart, the so-called M cells (PPK25-negative/*vGlut*-negative), which are responsive to male pheromones, inhibit this behavior. Both cell populations represent genetically and anatomically distinct neuronal classes, which enables the authors to study and manipulate them separately from each other. By using highly sophisticated genetic tools, the authors dissect the neuronal pathway of female pheromone processing. They convincingly show that F cells excite P1 neurons in the protocerebrum via PPN1 neurons, which represent pheromone projection neurons that provide the link between F cells in the VNC and P1 neurons in the higher brain to trigger courtship in the end. In addition, the authors study the neuronal pathway underlying male pheromone processing and elucidate the circuitry up to the higher brain. They provide evidence that M cells inhibit P1 neurons via activation of GABAergic mAL neurons in the protocerebrum which results in courtship inhibition. The authors conclude that the balance of an excitatory and inhibitory input onto P1 neurons depends on the activation of F and M cells and therefore contributes to courtship decisions. To their credit, the authors offer rational interpretations that are nuanced with appropriate caveats about alternative explanations and scenarios. In conclusion, this is a very beautiful and impressive study which is comprehensive and convincing. It represents a lot of work and is a welcome addition to the literature.

A few minor concerns that need to be taken care of to improve the current manuscript are enumerated below.

1) Figure 2: Why is the evoked calcium response significantly higher in P1 neurons when both the male and female pheromones are applied simultaneously in comparison to the application of the female pheromones alone? The authors argue that F cells are the only neurons that activate P1 neurons. However, if this assumption is true, one would not expect a synergistic effect. The authors should attempt to explain this effect and, if necessary, nuance their model accordingly.

2) Figure 3: A double labeling of PPN1 neurons with *ppk25-Gal4* or *vGlut-Gal4* is missing to label F cells selectively. Since the *ppk23-Gal4* lines labels both, M and F cells, it is not clear whether F cells indeed have contacts with PPN1 neurons. In addition, the authors should identify and name the brain area where PPN1 and P1 neurons overlap using the nomenclature published by Ito et al. (2014; Neuron).

3) Figure 5: The authors write in the figure legend that they monitored the calcium response following artificial activation of M cells via P2X2 expression, while they say in the manuscript that F+M cells were stimulated (subheading “M cells and F cells activate mAL neurons”). Since the authors used the *ppk23-LexA* line, they indeed stimulated F+M cells. Please modify the figure legend accordingly.

4) Figure 6: Please identify and label the brain area as mentioned above.

5) Please show a complete scheme of the neuronal circuitry at the end of the manuscript including all known neurotransmitter profiles (GABA, vGlut) and ion channels (PPK23, PPK25) to illustrate and simplify the final outcome of the study. Ideally, the anatomical location of the neurons (leg, VNC and brain) should also be indicated in such a diagram.

6) Paragraph three, Introduction, possibly needs a grammatical correction – "one set of neurons that has emerged… is the P1 neurons." – maybe change to "is the group of P1 neurons”?

7. Paragraph one of the Results states that "only the F neurons express the vesicular glutamate transporter." These are, in fact specifically labeled by the VGLUT-Gal4 line, which may not be quite the same thing as what is stated.

---

## [Author Response]

*1) Figure 2: Why is the evoked calcium response significantly higher in P1 neurons when both the male and female pheromones are applied simultaneously in comparison to the application of the female pheromones alone? The authors argue that F cells are the only neurons that activate P1 neurons. However, if this assumption is true, one would not expect a synergistic effect. The authors should attempt to explain this effect and, if necessary, nuance their model accordingly.*

We suspect that differences in *ppk23-LexA* and *vGlut-LexA* expression levels may account for the difference. We clarified this in the figure legend.

*2) Figure 3: A double labeling of PPN1 neurons with* ppk25-Gal4 *or* vGlut-Gal4 *is missing to label F cells selectively. Since the* ppk23-Gal4 *lines labels both, M and F cells, it is not clear whether F cells indeed have contacts with PPN1 neurons. In addition, the authors should identify and name the brain area where PPN1 and P1 neurons overlap using the nomenclature published by Ito et al. (2014; Neuron).*

We have provided a double label with PPN1-LexA and PPK25-Gal4 (Figure 3) as requested and used the recommended nomenclature.

*3) Figure 5: The authors write in the figure legend that they monitored the calcium response following artificial activation of M cells via P2X2 expression, while they say in the manuscript that F+M cells were stimulated (subheading “M cells and F cells activate mAL neurons”). Since the authors used the* ppk23-LexA *line, they indeed stimulated F+M cells. Please modify the figure legend accordingly.*

Legend changed.

*4) Figure 6: Please identify and label the brain area as mentioned above.*

This is modified in the legend.

5) Please show a complete scheme of the neuronal circuitry at the end of the manuscript including all known neurotransmitter profiles (GABA, vGlut) and ion channels (PPK23, PPK25) to illustrate and simplify the final outcome of the study. Ideally, the anatomical location of the neurons (leg, VNC and brain) should also be indicated in such a diagram.

We provided a schematic summarizing the results of the study (Figure 7).

6) Paragraph three, Introduction, possibly needs a grammatical correction – "one set of neurons that has emerged… is the P1 neurons." – maybe change to "is the group of P1 neurons”?

Changed as suggested.

7. Paragraph one of the Results states that "only the F neurons express the vesicular glutamate transporter." These are, in fact specifically labeled by the VGLUT-Gal4 line, which may not be quite the same thing as what is stated.

We changed the sentence to “F cells are the only leg neurons that express the *vesicular glutamate transporter-Gal4* driver (*vGlut-Gal4*).”